# Solid Dispersions Incorporated into PVP Films for the Controlled Release of Trans-Resveratrol: Development, Physicochemical and In Vitro Characterizations and In Vivo Cutaneous Anti-Inflammatory Evaluation

**DOI:** 10.3390/pharmaceutics14061149

**Published:** 2022-05-27

**Authors:** Bruno Vincenzo Fiod Riccio, André Luiz Carneiro Soares do Nascimento, Andréia Bagliotti Meneguin, Camila Fernanda Rodero, Kaio Pini Santos, Rafael Miguel Sábio, Sarah Raquel de Annunzio, Carla Raquel Fontana, Hernane da Silva Barud, Priscileila Colerato Ferrari, Marlus Chorilli

**Affiliations:** 1Department of Drugs and Medicines, São Paulo State University, Araraquara-Jaú Hwy. Km 1, Machados, Araraquara 14800-901, SP, Brazil; nascimento.a.l.c@gmail.com (A.L.C.S.d.N.); abagliottim@hotmail.com (A.B.M.); camilafrodero@hotmail.com (C.F.R.); kaiopsantos84@gmail.com (K.P.S.); rafaelmsabio@gmail.com (R.M.S.); 2Department of Clinical Analysis, São Paulo State University, Araraquara-Jaú Hwy. Km 1, Machados, Araraquara 14800-901, SP, Brazil; sarinha_annunzio@hotmail.com (S.R.d.A.); carla.fontana@unesp.br (C.R.F.); 3Department of Biotechnology, University of Araraquara, Carlos Gomes St., 1338, Centro, Araraquara 14801-320, SP, Brazil; hernane.barud@gmail.com; 4Department of Pharmaceutical Sciences, Ponta Grossa State University, General Carlos Cavalcantti Av., 4748, Uvaranas, Ponta Grossa 84030-900, PR, Brazil; priscileila@hotmail.com

**Keywords:** amorphization, anti-inflammatory activity, antimicrobial effect, crystallinity, technological innovation, skin permeation, solubility

## Abstract

*Trans*-resveratrol can promote various dermatological effects. However, its high crystallinity decreases its solubility and bioavailability. Therefore, solid dispersions have been developed to promote its amorphization; even so, they present as powders, making cutaneous controlled drug delivery unfeasible and an alternative necessary for their incorporation into other systems. Thus, polyvinylpyrrolidone (PVP) films were chosen with the aim of developing a controlled delivery system to treat inflammation and bacterial infections associated with atopic dermatitis. Four formulations were developed: two with solid dispersions (and *trans*-resveratrol) and two as controls. The films presented with uniformity, as well as bioadhesive and good barrier properties. X-ray diffraction showed that *trans*-resveratrol did not recrystallize. Fourier-transform infrared spectroscopy (FT-IR) and thermal analysis evidenced good chemical compatibilities. The in vitro release assay showed release values from 82.27 ± 2.60 to 92.81 ± 2.50% (being a prolonged release). In the in vitro retention assay, *trans*-resveratrol was retained in the skin, over 24 h, from 42.88 to 53.28%. They also had low cytotoxicity over fibroblasts. The in vivo assay showed a reduction in inflammation up to 66%. The films also avoided *Staphylococcus aureus*’s growth, which worsens atopic dermatitis. According to the results, the developed system is suitable for drug delivery and capable of simultaneously treating inflammation and infections related to atopic dermatitis.

## 1. Introduction

*Trans*-resveratrol is a phenolic compound and a phytoalexin found in natural sources, including grapes, peanuts, blueberries, and cranberries. It has low toxicity and, when topically administered, can be used as an anti-inflammatory, antioxidative, and antimicrobial agent [1,2,3]. As an anti-inflammatory agent, *trans*-resveratrol can modulate various enzymes, including kinases, lipoxygenases, cyclooxygenases, and sirtuins [4,5].

However, it possesses low solubility and high crystallinity, presenting issues for its incorporation into dosage forms and in its release, resulting in low bioavailability [6]. Some strategies enable its incorporation into pharmaceutical systems and increase its pharmacological effect, including amorphous solid dispersions [7].

In amorphous solid dispersions, the crystalline drug is dispersed into an inert and hydrophilic polymeric carrier that amorphized the drug or disperses its molecules into the molecular level inside the polymeric chains [8,9]. Thus, they can increase drug solubility in aqueous solutions, especially when surfactants are added to the formulation [10].

However, they alone cannot sustain controlled delivery to the skin since they generally lack the consistency required to remain in touch with the site of administration long enough to promote the drug’s release and achieve the required pharmacotherapeutic effect. Therefore, their incorporation into other systems can overcome this issue if these systems are able to maintain the contact with the target tissue [11].

Polymeric films are drug delivery systems that enable drug vectorization to the targeted tissue, such as the skin [12,13]. They can be used to release the drug to the skin via the diffusion process through a drug gradient between the interior of the film and the exterior (the skin), causing controlled drug release and improving treatment [14].

Polymeric films are ideal for drug loading to treat inflammatory skin diseases such as dermatitis since they can be administered over the affected skin [15]. In addition, they are advantageous over semi-solid conventional formulations because they require a lower frequency of administration. In addition, they promote skin dressing protecting the injured skin from friction with clothes and forms a protection against other environmental aggressors [16].

This study aimed to develop and characterize PVP polymeric films containing third-generation amorphous solid dispersions of *trans*-resveratrol dispersed into chitosan and D-α-tocopherol polyethylene glycol 1000 succinate (a.k.a. TPGS) as a drug delivery system to treat inflammatory skin diseases.

The film’s characterization regarded their barrier properties and mechanical aspects and characterized their morphological and physicochemical properties by SEM, XRD, FT-IR, and thermal analysis (TGA and DSC). The films were evaluated in vitro, such as their bioadhesion, drug release profile, drug permeation, and cytotoxicity in cellular fibroblasts cells; the in vivo efficacy was analyzed using the mouse ear edema model, the disk diffusion assay was used to evaluate the antimicrobial effect against *S. aureus*.

## 2. Materials and Methods

### 2.1. Materials

TPGS (Sigma-Aldrich, Saint Louis, MO, USA), Ethyl alcohol (Êxodo Científica, São Paulo, Brazil), Glacial acetic acid (J.T. Backer, São Paulo, Brazil), Glicerine (Synth), Low molecular mass chitosan (poly-D-glucosamine) (Sigma-Aldrich, Saint Louis, MO, USA), Poloxamer^®^ 407 (Sigma-Aldrich, Saint Louis, MO, USA), Polyethylenoglycol 400 (Synth), Purified Water, PVP 360 (Sigma-Aldrich, Saint Louis, MO, USA), Resveratrol extract 100% (Galena), Sodium benzoate (Pantec, Kradolf, Switzerland).

### 2.2. Methods

#### 2.2.1. Solid Dispersions’ Manufacture

Solid dispersions were composed of chitosan, *trans*-resveratrol, and TPGS, varying the drug: polymer ratio (Table 1). They were prepared by solubilizing chitosan in 15% acetic acid aqueous solutions under mechanical stirring at 700 rpm for 1 h. Then, the drug was pre-solubilized in ethylic alcohol and incorporated into the polymeric solution, and the TPGS powder was added and stirred under the same conditions.

After the homogenization, the material was dried in a spray dryer MSD 1.0^®^ (LabMaq, São Paulo, Brazil). The drying was carried out under the following conditions: −50 mm Hg, the outlet temperature of 120 °C, blower of 1.5, feeding rate of 0.3 L/h. The obtained powders were stored away from light, heat, and humidity.

#### 2.2.2. PVP Films Preparation

PVP films formulations are also presented in Table 1. Solid dispersions (or raw chitosan in control samples) and TPGS were solubilized in a 1:1 water and ethanol mixture, and the final preparation was acidified with 1% (*v*/*v*) acetic acid. The polymeric suspensions were stirred at 700 rpm for 1 h, and then 1% (*w*/*w*) PVP, 0.1%, sodium benzoate, 0.05%, glycerin, and 0.05% polyethylene glycol (PEG 400) were added and stirred for 1 h. The suspensions were poured into Petri dishes (30 g) and dried in a drying oven at 40 °C for 12 h. Finally, the obtained films were stored out of heat and light. 

### 2.3. Liquid Uptake Ability

Liquid uptake of films was studied using an adapted Enslin funnel (Meneguin, et al. [17]). Briefly, membrane sections were placed on the sintered glass plate of the funnel, which was attached to a horizontally arranged graduated pipette at the same height as the funnel through a silicone hose. The entire set was filled with PBS pH 7.4 and closed with a parafilm aid. At predetermined times, the absorbed liquid volume was registered and used to calculate the liquid uptake according to Equation (1):(1)$$\mathrm{Liquid}\;\mathrm{uptake}\;\mathrm{ability}=\frac{\mathrm{Absorbed}\;\mathrm{liquid}\;\mathrm{volume}\;\left(\mathrm{mL}\right)}{\mathrm{Mass}\;\mathrm{of}\;\mathrm{dry}\;\mathrm{sample}\;\left(g\right)}\times 100$$

Equation (1): Liquid uptake ability from the PVP films.

### 2.4. Water Vapor Permeability (WVP)

The WVP of the membranes was gravimetrically evaluated as described by Akhagari, et al. [18]. Briefly, circular sections of each sample were fixed between the top of the glass cups and caps with a 1.1 cm opening. The glass cups were filled with 10 mL of water (100% relative humidity; RH). The set was weighed and placed in a desiccator containing silica gel (0% RH). The cups were removed from the desiccator and weighted at predetermined times (24, 48, 72, 96, and 120 h), and the weight loss was used to calculate the *WVP* (Equation (2)):(2)$$WVP=\frac{WVTR.L}{A.P_{0}.\left(RH_{1}-RH_{2}\right)}$$

Equation (2): Water vapor permeability from the PVP films.

Where:

*WVTR* = water vapor transmission rate (calculated from the slope of weight loss versus time curves, g m^−2^ h^−1^);

*L* = average film thickness (mm);

*A* = surface area of the film exposed to the vapor (m^2^);

*P*_0_ = pure water vapor pressure (3.159 kPa at 25 °C);

*RH*_1_ and *RH*_2_ = the relative humidity gradient.

### 2.5. Mechanical Properties

The evaluation of the mechanical properties of the films was performed using a texture analyzer TA-XT2 (MicroSystems) equipped with a spherical-ended puncture probe (5 mm) and with the membranes sections fixed on a metallic holder (10 mm opening). The probe moved down at 0.10 mm⋅s^−1^ with a trigger force of 0.005 kg during the test. The force vs. displacement curves were recorded and used to estimate the puncture strength (*P_s_*), elongation at break (*E_b_*), and perforation energy (*P_E_*), according to Equations (3)–(5) [17]:(3)$$P_{s}=\frac{F}{A}$$

Equation (3): Puncture strength of the PVP films.

Where:

*F* (N) = the maximum force to rupture the film;

*A* (m^2^) = sectional area of the film ((calculated from *A* = 2*rh*), where *r* is the hole radius and *h* is the thickness).
(4)$$E_{b}=\frac{\sqrt{r^{2}+d^{2}}-r}{r}\times 100$$

Equation (4): Elongation at the break of the PVP films.

Where:

*r* (mm) = radius of the exposed film on the plate orifice;

*d* = probe displacement.
(5)$$P_{E}=\frac{AUC}{V}$$

Equation (5): Perforation energy of the PVP films.

Where:

*AUC* = area under the curve force vs. displacement;

*V* = film’s volume (calculated from *V* = *πr*^2^*h*, where *r* is the hole radius and *h* is the film’s thickness).

### 2.6. Scanning Electron Microscopy (SEM)

The surface and cross-sections of the films were examined (JEOL JSM 7500F; Peabody, MA, USA). A carbon film was added onto the stubs before analysis, and then the samples were deposed. After that, the samples were sputter-coated with a thin carbon layer to make them conductive materials. The photomicrography of the surface and cross-sections of the sample were acquired at 35.000 and 300×, respectively [19].

### 2.7. X-ray Diffraction (DRX)

The X-ray patterns were obtained using a Siemens D5000 X-ray diffractometer with CuK radiation (λ = 1.541Å) and a setting of 40 kV and 40 mA. The diffractograms were obtained under 2θ open-angle X-ray scanning from 2 to 40° and a goniometer velocity of 0.05%. Physical mixtures between the formulations’ main components were prepared in a 1:1 (*w*/*w*) ratio to evaluate possible changes in the crystalline profile of *trans*-resveratrol [20].

### 2.8. Fourier Transformed Infrared Spectroscopy (FT-IR)

FT-IR spectra were acquired on a VERTEX^®^ 70 Bruker (DLaTGS detector; Billerica, MA, USA) spectrometer in the range from 4000 to 400 cm^−1^ employing 64 scans (resolution of 4 cm^−1^) and the attenuated total reflectance mode (ATR) [21].

### 2.9. Thermogravimetry and Differential Scanning Calorimetry (TG-DSC)

Simultaneous thermogravimetry and differential scanning calorimetry (TG-DSC) thermograms were obtained using a TG-DSC1 STARe system (Mettler Toledo, Columbus, OH, USA). The dry air was used as a purge gas (flow rate of 50 mL⋅min^−1^). Alumina crucibles were used to place the samples. They were heated at a rate of 10 °C min^−1^ until 335 °C, except for PVP (up to 655 °C).

### 2.10. In Vitro Bioadhesion

The texture analyzer A-XT Plus^®^ (Extralab, Itatiba, Brazil) was used in the adhesion test mode. Porcine ear skin (Frigodeliss, São Paulo, Brazil) was dermatomized into 500 µm with a dermatometer TCM^®^ 300 (Nouvag, Goldach, Swiss), hydrated with a solution of 0.9% NaCl for 15 min at 32 °C, and fixed in the probe of the equipment. The film fragments (1 cm²) were fixed over an acrylic plate. The probe went down at 1 mm/s^−1^ constant speed until it encountered the film, keeping in contact with a strength of 0.05 N for 300 s. The skin was separated from the film at 1 mm/s^−1^ and the separation resistance force was registered. A curve strength versus time was generated, and the bioadhesion strength (N) was determined [22].

### 2.11. In Vitro Trans-Resveratrol Release

The films were cut in a standardized way (n = 6) and placed in the donor compartment with an exposure area of 1.77 cm^2^, and the receptor medium was filled with phosphate buffer (pH 7.4) and sodium lauryl ether sulfate. The experiments were performed at 32 ± 0.5 °C and 300 rpm, ensuring the sink condition. A total of 2 mL aliquots were collected at 30 min, 1, 2, 4, 8, 12, 16, 20, 24 h to assess the release kinetics [23]. The quantification of *trans*-resveratrol release was measured via HPLC using validated analytical conditions.

### 2.12. In Vitro Permeation

The porcine skin used for permeation was cleaned and dermatomized into 500 µm with a dermatometer TCM^®^ 300 (Nouvag, Goldach, Swiss) before the assay. The porcine skin was placed in the donor compartment, and the film was placed above it. The equipment and parameters were the same for the in vitro drug release study.

After the permeation assay, the porcine skins were removed from the Franz cells and separated from the films. The skins were washed with distilled water and dried with absorbent paper. Immediately following the drying, tape stripping was performed to remove the *corneum stratum* using 16 adhesive tapes (Scotch 750 3M), discarding the first. The tapes were placed in test tubes with 5 mL acetonitrile and vortex agitated for 1 min, then the solutions were submitted to an ultrasonic bath for 15 min. The solutions were filtered in syringe filters (0.45 µm), and the drug was quantified by HPLC.

Drug retention was also evaluated. The skins were cut into small fragments and added to the test tubes with 5 mL of acetonitrile, which was homogenized in a vortex for 2 min, followed by Ultra Turrax for 1 min, and an ultrasonic bath for 30 min. The solutions were filtered (0.45 µm), and the drug was quantified by HPLC.

### 2.13. In Vitro Cytotoxicity

The study was performed according to other studies published by our research group [3,24]. It was carried out using L929 cells (fibroblast). The cells were cultured in T-flasks (75 cm^2^) using Dulbecco’s Modified Eagle’s Medium. The culture medium was supplemented with fetal bovine serum (5%, *v*/*v*) and antibiotic solution (1%, *v*/*v*) containing 10,000 UI of penicillin and streptomycin 10 mg⋅mL^−1^. Cells were incubated at 5% CO_2_ and 37 °C in a humidified atmosphere. At about 80% of confluence, the cells were detached using trypsin (0.05%, *v*/*v*), and 8 × 10^5^ cells were seeded into 6-well microplates.

The plates remained incubated at 37 °C with 5% CO_2_ for 48 h to allow cell adhesion and 75–80% confluence. After 48 h, the culture medium was removed, and the wells were washed with PBS. Then, 1 mL of the culture medium containing agar (0.9%, *w*/*v*) and neutral red dye (0.01%, *w*/*v*) was added to each well. After agar solidification, standardized PVP film fragments were placed in the center of the agar-containing well. A sterile paper disk was embedded in a *trans*-resveratrol (3 mg/mL) suspension in 0.5% DMSO (*w*/*v*). A filter paper disk containing only culture medium was used as a negative control, while a disk containing Triton-X was used as a positive control. The plates were wrapped in aluminum foil to avoid cell damage by photoactivation of neutral red and placed in an oven at 37 °C with 5% CO_2_ for 24 h.

After incubation, the wells were macroscopically observed, and the formation of a clear halo was measured with a pachymeter. The study was performed in triplicate and expressed according to the ISO 10993-5:2009 (Table 2).

### 2.14. In Vivo Anti-Inflammatory Activity

For the in vivo study, male Swiss albino mice (*Mus musculus*) weighing 25–30 g were used. The experiment was approved by the Ethics Committee for Use of Animals (Comissão de Ética no Uso de Animais da Faculdade de Ciências Farmacêuticas de Araraquara), UNESP, under the protocol CEUA/FCF/CAr number 31/2020. The animals were from the UNESP Central Vivarium, and the experiment occurred in the UNESP Drugs and Medicines Department vivarium. The animals were maintained in polypropylene boxes and under controlled environmental conditions: 23 ± 1 °C, 55 ± 5% RH, and light cycles of 12 h. The water and food supply were ad libitum (with no restrictions). The animals were anesthetized with an injection of 100 mg/Kg ketamine hydrochloride (Cetamin Syntec^®^, São Paulo, Brazil) and 10 mg/Kg xylazine chloridate (Sipeagro^®^, São Paulo, Brazil) via intraperitoneal administration. 

The inflammatory process was induced by administering 20 µL croton oil (2.5% *v*/*v*) diluted in acetone in the internal part of the animal’s ears (1 cm² approximately) [25]. Seven experimental groups of 6 animals each (n = 42) were used, including a *trans*-resveratrol suspension (3 mg/mL) with 0.2% (*w*/*v*) sodium lauryl sulfate, the films samples (test and control), a positive control dexamethasone commercial cream 10 g (1 mg/g) from EMS Pharmaceuticals, and a negative control (no treatment). The treatment consisted of applying 20 mg of formulation into the animal’s right ear.

Six hours after the treatment, the mice were euthanized with carbon dioxide (CO_2_) by inhalation. Then, both ears of each mouse were cut using a dermatological punch and weighed. The anti-inflammatory activity was expressed according to the percentage of edema reduction by comparing the difference in weight of both ears, and the groups were compared by ANOVA test (*p* < 0.05) [26].

### 2.15. In Vitro Antimicrobial Activity Evaluation against Staphylococcus aureus

#### 2.15.1. Bacterial Strain

An *S. aureus* ATCC 25923 strain was utilized (from Instituto Nacional de Controle de Qualidade em Saúde, Oswaldo Cruz Foundation—FIOCRUZ. Rio de Janeiro, Brazil).

#### 2.15.2. Culture Media and Bacterial Strain Reactivation

The bacterial strain was reactivated in a solid medium of Trypticase Soy Agar (TSA; Kasvi^®^, Curitiba, Brazil). A vial containing the strain was kept frozen at −80 °C and thawed at the moment of the reactivation. A 20 μL aliquot was sown over the medium by the streaking method. Afterward, the petri dishes were incubated at 37 °C for 24 h [27].

#### 2.15.3. Disk Diffusion Assay

The antimicrobial activity of the polymeric films was studied by the disk diffusion assay adapted from the standard protocol of the Clinical and Laboratory Standards Institute (CLSI) [28]. An *S. aureus* strain inoculum in Muller-Hinton broth was made with the cellular concentration adjusted in a spectrophotometer to 1 × 10^8^ cells/mL^−1^, approximately. The bacterial suspension was scattered over the surface of the Muller-Hinton agar using sterile swabs. Then, the four formulations of polymeric films (diameter = 19 mm, 9/group) were inserted onto the agar using sterile tweezers, and the dishes were incubated in aerobiosis at 37 °C for 24 h. After the incubation, the formed halos were measured.

As positive controls, filter paper disks (19 mm, 9/group) containing *trans*-resveratrol solubilized in ethanol were divided into two groups, one with the concentration corresponding to the concentration of the drugs’ membranes. The negative controls comprised of only ethanol incorporated into filter papers, and without ethanol or *trans*-resveratrol.

### 2.16. Statistical Analysis

The mean values ± standard deviations were carried out in the evaluations. An ANOVA test and Tuckey post-test were used to compare the groups. Significant differences were considered for *p* ≤ 0.05 [29].

## 3. Results

### 3.1. PVP Films Preparation

Solid dispersions were successfully obtained, showing a yellow-brown appearance. The films presented uniform aspects, malleability, and brilliant and smooth surfaces. However, films containing solid dispersions were more opaque than the control films.

The thickness of the films was measured, and the results are presented in Table 2. The control samples showed significantly higher values than films prepared with solid dispersions.

### 3.2. Liquid Uptake Ability

Liquid uptake at equilibrium was observed after 120 min. The developed films showed high liquid uptake ability, ranging from 606.69% to 1579.59% (Table 2).

The control sample C-PVP-1:9 showed higher liquid uptake (1579.59%) than the solid dispersion film (PVP-SD-1:9).

### 3.3. Water Vapor Permeability (WVP)

The films prepared with solid dispersions presented lower WVP than their respective controls: from 1.38 to 1.43 for the ones with solid dispersions and from 2.07 to 2.10 for the controls (Table 3).

### 3.4. Mechanical Properties

The formulation PVP-SD-1:3 was approximately 70% stronger than the respective control (Ps values of 39.25 ± 2.67 Mpa vs. 22.99 ± 5.37 Mpa, respectively; *p* < 0.05) (Table 3).

### 3.5. Scanning Electron Microscope (SEM)

The SEM images (Figure 1) show films with uniform and smooth surfaces without pores, whereas the PVP-SD-1:3 showed a slightly irregular surface (Figure 1G) at a nanometric scale.

### 3.6. Fourier-Transformed Infrared Spectroscopy (FT-IR)

Figure 2A exhibits the films, *trans*-resveratrol, PVP, and chitosan FT-IR spectra. Most of the bands in the film spectra displayed similar profiles, with bands ascribed to the matrices more intense than any isolated polymer or drug.

### 3.7. X-ray Diffraction (XRD)

The X-ray diffraction patterns are displayed in Figure 2B. *Trans*-resveratrol has well-defined and narrow peaks, while chitosan has an amorphous structure profile, having an extensive and poorly defined halo around 10° and another between 18° and 23°. PVP showed only two broad characteristic peaks at 11° and 21°.

The physical mixtures *trans*-resveratrol: PVP (1:1) and *trans*-resveratrol: chitosan (1:1) showed a similar profile to the pure drug, while the chitosan: PVP (1:1) showed a profile more like the raw chitosan. Finally, all membranes show only traces of the semi-crystalline structure of chitosan and PVP.

### 3.8. Thermal Analysis (TG and DSC)

The thermal data from the TG and DSC curves were summarized in Table 4. The TG and DSC curves are presented in Figure 3.

### 3.9. In Vitro Bioadhesion

The bioadhesion strength of the films in porcine skins is presented in Table 5.

### 3.10. In Vitro Release of Trans-Resveratrol

The total drug release from the films in 24 h was 244.35 ± 8.07 μg/cm^2^ (82.27%) for the PVP-SD-1:9 and 745.67 ± 32.84 μg/cm^2^ (92.81%) for the PVP-SD-1:3, respectively.

Figure 4A shows the in vitro *trans*-resveratrol release from the film’s profile. It was observed as an extended-release during 24 h, although the totality of the drug was not released at this time. At the first hour of the experiment, less than 5% of the drug was released from the films [30], which may be related to the swelling process beginning after exposition to the receptor medium.

### 3.11. In Vitro Cutaneous Permeation

When comparing the amount of drug found in the receptor medium with the total found by combining the tape stripping, cutaneous retention, and cutaneous permeation, 92.04 ± 8.72 μg/cm² (PVP-SD-1:9) and 243.05 ± 11.37 μg/cm² of skin (PVP-SD-1:3) was found in the receptor medium, representing 30.68 ± 2.91 and 31.22 ± 1.46% of the total drug content in the three experiments. The total permeated drug (combining the three experiments) from PVP-SD-1:9 was 234.70 μg/cm² of skin (83.88%), while 711.73 μg/cm² of skin (91.25%) *trans*-resveratrol from PVP-SD-1:3 film. The cutaneous permeation can be seen in Figure 4B and the drug retention in Table 6.

### 3.12. In Vitro Cytotoxicity

The in vitro cytotoxicity was evaluated after the 24 h samples’ incubation, where the film’s medium discoloration and halo appearance were checked, indicative of fibroblast viability decrease, as shown in Table 7.

### 3.13. In Vivo Anti-Inflammatory Activity

Table 8 shows the percentages of inflammation and inflammation reduction values obtained in the in vivo assay. The *trans*-resveratrol suspension and the membranes showed the highest values of inflammation reduction (after the positive control), proving the anti-inflammatory action and demonstrating that its incorporation into membranes maintains this activity. Still, it could be noted that the untreated group (negative control) and membranes (without drug) showed similar values among them, indicating that the anti-inflammatory action is due to *trans*-resveratrol.

### 3.14. In Vitro Antimicrobial Effect against Staphylococcus aureus—Disk Diffusion Assay

With this assay, it was possible to observe the inhibition of the growth of *S. aureus* around the membrane, and there was no inhibition of bacterial growth when the *trans*-resveratrol solution in ethanol (125 µg) was incorporated into sterile filter paper disks.

Table 9 shows the mean values and the standard deviations of the inhibition halo values in mm. From these values, it was found that only the PVP-SD-1:3 membrane showed inhibition of the growth of *S. aureus*, while other tests groups showed no inhibitory effect.

## 4. Discussion

The films were successfully developed and showed favorable characteristics for their purpose. The uniformity can assure the covering of the affected area and that the drug delivery will be uniform, providing more safety and efficacy during treatment, assuming that the drug is homogeneously dispersed through the formulation and has an appropriate release tax [31].

The chitosan present in the solid dispersions (PVP-SD-1:9 and PVP-SD-1:3) has faster solubilization than its raw state, probably because the chitosan is protonated inside of the solid dispersions. Since *trans*-resveratrol is amorphized, it solubilizes faster and stays stable for longer.

Cherukuri, et al. [32] reported that the thickness of polymeric films could be related to the solubility of these polymers in the system solution before the drying, resulting in higher and faster solubilization and producing thinner films. Another critical factor is that solubilized chitosan is available to interact with the other components, including the PVP, causing polymeric chains compaction, turning films thinner and smooth (without clusters).

The observation of the liquid uptake ability behavior was expected because PVP and chitosan can swell in an aqueous medium due to their high hydrophilicity. The polymeric chains of the hydrophilic polymers tend to form hydrogen bonds with water molecules when in an aqueous medium, or even capture the humidity of the environment, causing the swelling of the polymeric network and, therefore, increasing the size and weight of the material [33].

The uptake of the control membranes compared to their counterparts having solid dispersions happened due to the higher proportion of free hydroxyl groups in the chitosan, which is not interacting with *trans*-resveratrol. It is worthy to emphasize that acetic acid in the production of solid dispersions and films promotes the ionization of chitosan, causing polymeric chains’ electrostatic repulsion and increasing the formulation’s hydrophilicity. In this sense, the swelling also varies according to the polymer concentration and the nature of the intermolecular interactions that they can establish.

Additionally, the thickness of the polymeric films can affect the swelling phenomenon due to the surface area in contact with the biological fluids. For example, thinner films have a proportionally higher surface area that allows a more significant proportion of polymer to come into contact with the liquid, hydration to occur faster, swelling, and release when compared with thicker films [34].

PVP and chitosan demonstrated high water vapor permeability, related to the rise of the permeability coefficient, and swelling under high relative humidity due to adsorption isotherm phenomena [35].

The lower WVP of the films containing solid dispersions than the control films can be related to the *trans*-resveratrol in formulations building a more robust and structured drug-polymeric network. In addition, the drug possesses a hydrophobic character, and the interaction with water molecules adsorbed on the surface of the membranes is reduced [36]. Besides, the films showing the lowest WVP values also show the lowest liquid uptake ability. Such behavior can be related to the fact that swelling is responsible for the increased system dimensions due to polymeric network expansion, generating large empty spaces between the polymeric chains, which facilitate the water vapor permeation [35,37].

Knowing WVP properties provides essential insights into the formed film’s various physicochemical phenomena and structural factors. The proper balance between the polymeric characteristics and the entanglement degree of the polymeric network is pivotal for the excellent barrier property, which will interfere with the permeability of water vapor and gas exchange with the tissues, as well as the degree of drug entrapment into the empty spaces and its consequent release. In other words, a permeability study allows us to understand the mass transfer mechanisms and the polymer–polymer and polymer–drug interactions. Several factors impact the WVP, such as the material nature (hydrophilic or hydrophobic), the pore’s presence, and the film tortuosity [38].

The mechanical properties suggest that a solid dispersion establishes supramolecular drug–drug, drug–polymer, and polymer–polymer interactions, which are responsible for a higher resistance threshold before the yield point is reached and mechanical perforation occurs.

Felton, et al. [39] and Meneguin, et al. [40] have stated that less concentrated polymeric films generally have greater resistance against rupture. A looser network with larger intermolecular spaces allows a conformational rearrangement until the yield stress is reached. Although PVP concentration is the same for both samples, there are different amounts of chitosan in the films, showing that the highest proportion of chitosan had the highest *P_s_*, despite no statistically significant differences occurring. Corroborating with the WVP and liquid uptake results, the PVP films produced with solid dispersions had a lower WVP, liquid uptake, and higher mechanical strength than the control films due to the more entangled drug: polymer network.

The SEM cross-section images exhibited a uniform aspect that can indicate a homogeneous drug dispersion in the film. In addition, uniformity is essential to maintain the coverture of the affected skin area, as discussed before. The good covering causes the homogeneity of the drug delivery, and the absence of pores of ruptures leads to a lower WVP, maintaining humidity and entrapping exudates and allowing gas exchange to occur, both necessary to accelerate skin healing [41].

Regarding the FT-IR analysis, bands at 2923, 1460, and 1374 cm^−1^ were assigned to the –CH2 and –CH stretching in the film samples. At 1285 cm^−1^, the band corresponds to the –C-N bound from PVP, whereas carbonyl (–C=O) groups were observed at 1650 cm^−1^ [42]. Karavas, et al. [43] reported that hydrogen bonds could favor interactions between PVP and chitosan. The hydroxyl groups present in chitosan are electron donors, being interesting groups to establish hydrogen bound with carbonyl groups of PVP, ensuring that both of the precursors are making interactions showing their good chemical compatibility.

*Trans*-resveratrol showed bands between 3500 and 3200 cm^−1^, related to the phenolic hydroxyl groups, which can shift by interaction with PVP methyl, indicating hydrogen bonds formation [44,45]. It is highlighted that PVP-SD-1:3, which contains more amounts of the drug, at 840 cm^−1^, exhibited a more intense band than PVP-SD-1:9, which is attributed to the =C-H stretching of arene conjugated to the olefinic group [46], characteristic of *trans*-resveratrol.

In the XRD analysis, it is possible to affirm that the raw *trans*-resveratrol presented a highly crystalline profile since its diffractograms show well-defined peaks [47], unlike chitosan and PVP that are more like semi-crystalline structures, being polymers composed of an amorphous part and some crystalline structures along their chains [48,49].

The physical mixtures of *trans*-resveratrol: PVP (1:1) and *trans*-resveratrol: chitosan (1:1) indicated that the drug amorphization does not occur through a simple mixture with polymers, which means that the use of the employed technologies leads to the amorphization of the drug. It can be observed by the profile of the films containing the drug, demonstrating the effectiveness of using solid dispersions in *trans*-resveratrol amorphization.

By observing the thermal analysis, the DSC results showed that *trans*-resveratrol is stable until 242 °C, and the thermal decomposition happens in two consecutive and overlapping steps until 650 °C, corresponding to 95.95 % of mass loss. A sharp endothermic peak is observed at 268 °C, attributed to the melting of the material, since the drug was obtained in the crystalline state [50]. An exothermic peak can be seen at 565 °C, referring to the oxidation of organic matter. PVP showed the first mass loss step from 30 °C to 135 °C, with an endothermic event at 75 °C in the DSC thermogram, probably due to the dehydration of adsorbed water. The dry polymer is stable up to 240 °C, and the thermal decomposition happens through a slow process until 400 °C, followed by two more consecutive steps of mass loss, from 400 °C to 455 °C and from 455 °C to 650 °C, respectively. The final residue found is equivalent to 0.12%.

Chitosan dehydration was observed in a single step up to 129 °C, corresponding to 8.47% of mass loss, associated with an endothermic peak at 73 °C in the DSC curve. Above this temperature, the material presented low thermal stability, which decomposes from 149 °C until 310 °C, followed by the last step between 310 °C and 607 °C, both occurring through a slow process, associated with endo- and exothermic peaks in the DSC thermogram. PVP showed dehydration at 71 °C and an indication of an exothermic event close to 335 °C.

All films presented a similar mass loss profile in the TG analysis until the third step (Table 3). The first and second steps are consecutively associated with dehydration, followed by decomposition of the materials. The absence of the fifth step can be observed for PVP-SD-1:3, which can indicate a better amorphization in the PVP-SD-1:9 film, as crystalline compounds often show step mass losses (or lower thermal stability). From the calorimetric data, it is possible to observe that the endothermic peak of *trans*-resveratrol is absent in the films, suggesting its amorphization [50]. The films prepared with solid dispersions had similar thermal profiles to the control ones, suggesting no chemical incompatibilities.

The physical mixtures present thermal profiles similar to the pure compounds, indicating no incompatibility between the materials. It was possible to notice that the endothermic event related to the drug melting was significantly reduced and displaced in the mixture with chitosan, and it can hardly be seen in the mixture with PVP. The physical mixture between chitosan and PVP presents similarity with the raw chitosan, reducing both endo- and exothermic events. It may suggest that just drug and polymers as physical mixtures are insufficient to amorphize the *trans*-resveratrol.

In the bioadhesion, the samples containing more chitosan (1:9 drug: polymer ratio) had more adhesivity bonding force. The bioadhesive potential depends on the ability of film’s surface in contact with the skin to interact via hydrophobic and hydrophilic effects. The polymeric chains exposed to the surface can make hydrophobic bonds with wet soft tissues, anchoring themselves to the hydrophobic groups in the phospholipid membranes and the extracellular proteins, thus promoting a better adhesivity [51].

PVP shows bioadhesive properties in different animal tissues [52]. PVP has a hydrophobic part that interacts with the cells and proteins present on the skin’s surface, but it also has a hydrophilic part that starts to be more adhesive when the relative humidity is over 40% [53]. Pagano, et al. [54] stated that PVP has suitable bioadhesive properties due to its hydrophobic groups that interact with the surface’s proteins and other molecules present in the *stratum corneum* of the porcine ear skin that cause the bioadhesion.

Bioadhesion is a crucial characteristic used in dermatologic medicines, related to the length of stay of the formulation when in contact with the skin [55]. Our results indicate that the PVP membranes can maintain contact with the tissue to release the *trans*-resveratrol leading to a more significant action time, bioavailability, and effectiveness of the therapy.

The behavior seen in the in vitro release of *trans*-resveratrol could be related, mainly because PVP is a polymer able to sustain drug delivery. When used to form films, it can be loaded with the drug, maintain its stability, inhibit crystal growth, and be bioadhesive [56], all of which are essential for cutaneous drug release. Therefore, free *trans*-resveratrol (crystalline) has a first-order kinetics diffusion in different media which changes when amorphized [57].

Besides PVP, chitosan in solid dispersions also influenced drug delivery, demonstrating that the sample with a 1:9 drug: polymer ratio presents a more prolonged trans-resveratrol release. Chitosan is a polymer widely employed for controlling drug delivery, and the primary mechanism is due to its polymer swelling ability [58].

In the in vitro release, it is noteworthy to report that each film had a different permeation profile; the formulation containing more chitosan (1:9) showing a slower permeation, prolonging the drug release, as discussed before, and therefore, taking longer to concentrate in the receptor medium, indicating that the release profile also influences the permeation.

Since the *stratum corneum* is considered the main biological barrier to the permeation of drugs, it is essential to evaluate if the drug can surpass this layer. Since PVP inhibits the recrystallization of drugs, it is possible to relate the PVP concentration to the *trans*-resveratrol release and permeation rates in the skin [59]. Besides, skin retention and tape stripping are essential techniques for assessing the depth where the drug stays longer and are critical analyses to understand the drug’s partition at the site of its action.

The TPGS presence in solid dispersions can favor skin permeation since it causes interfacial tension reduction and modifies the stratum corneum’s selectivity, allowing the drug to permeate the skin through the intercellular, transcellular, or transcellular cutaneous appendage pathways [60].

The sample with a 1:9 drug: polymer ratio possesses a minor amount of drug in the formulation, and the PVP-SD-1:3 showed higher drug concentrations (µg/cm^2^) in the *stratum corneum* and dermis related to its higher drug release rate.

In this sense, the application of the developed films is promising in dermatitis treatment due to their ability to control *trans*-resveratrol release, and to be retained in the epidermis layer in high amounts.

Regarding the results seen in the cytotoxicity, as observed by Maccario and collaborators [61], *trans*-resveratrol has an antiproliferative effect in fibroblast cell cultures when in high concentrations, although it is not the usual observation. Anlar, et al. [62], when testing resveratrol in a fibroblasts lineage, observed that, in 24 h, 50 μL/mL or more were able to reduce cell viability, and 400 μL/mL decreased the cell viability to 55%. Therefore, the cytotoxicity data for *trans*-resveratrol agree with the literature’s findings.

However, the cytotoxicity results showed that cell death is not directly related to the concentration of *trans*-resveratrol available in the films. The film with the highest concentration of resveratrol (PVP-SD-1:3) showed minor cytotoxicity compared to the film with the lowest concentration of *trans*-resveratrol (PVP-SD-1:3). It is probably due to the presence of chitosan.

Although chitosan is biocompatible, it is already well-established that chitosan is an excellent permeation promoter [63]. Thus, the film with the highest chitosan concentration promotes greater *trans*-resveratrol permeation in the cell membrane, and it results in higher cytotoxicity in films with a lower concentration of the drug. Although the assay shows cell death, this rate is moderate and tolerable when dealing with a drug delivery system for topical use.

It is still important to highlight that films containing the drug have smaller halos than the pure drug, indicating that incorporating the drug in a controlled drug delivery system, such as the PVP films, reduces its cytotoxicity. Both films had a considerable decrease in toxicity, leading to a safer cutaneous administration than just the drug in suspension.

In the in vivo assay, the induced mouse ear edema is the most common animal and histological model for testing cutaneous administered drugs with topical anti-inflammatory activity [64]. Despite no statistical difference among the samples that contained the *trans*-resveratrol, the PVP-SD-1:3 formulation had a better therapeutic effect. It can be due to faster drug release at the beginning with this sample, which promoted a more evident inflammation reduction in the short treatment period of six hours.

Another point to be discussed is that the films can stay on the skin longer due to their bioadhesion. In this sense, the *trans*-resveratrol will be released longer, maintaining its action, indicating that the use of this system is advantageous compared to drug suspension, which possesses lower contact time with the tissue.

The evaluation of the antimicrobial action of the membranes was performed using the disk diffusion method. This method has been commonly used in studies evaluating solid drug delivery systems [65].

A strain of *S. aureus* was utilized since it is the most critical microorganism associated with the skin affected by atopic dermatitis [66]. Therefore, inhibiting the growth of this bacterium is pivotal to avoiding the aggravation of this disease and may accelerate skin healing.

The drug amorphization can explain this result in the films, which causes an increase in *trans*-resveratrol solubility, and its release is facilitated when the membranes are hydrated. As for the filter papers containing the *trans*-resveratrol in a solution of ethanol, the molecules are probably in the crystalline state (principally after the evaporation of ethanol), so the drug does not pass easily through the filter paper, and the release to the medium is complex, affecting the antimicrobial effect.

Furthermore, it is essential to mention that the activity was dependent on the *trans*-resveratrol concentration, considering that PVP-SD-1:9, containing 48 µg of *trans*-resveratrol, did not prevent the growth of the bacterium *S. aureus.* In contrast, PVP-SD-1:3, containing 125 µg of *trans*-resveratrol, could prevent it. This result is in accord with the work carried out by Vivero-Lopez, et al. [67], in which the authors found, in *S. aureus* and *Pseudomonas aeruginosa,* that the antibacterial effect of resveratrol was dependent on the concentration.

The mechanism by which *trans*-resveratrol prevents the growth of *S. aureus* still needs to be better understood [68,69]. However, it has been reported that *trans*-resveratrol can attenuate this bacterium’s virulence without affecting the pathogen’s survival. The mechanism of action is related to the inhibition of alpha-hemolysin expression through the downregulation of HLA transcription, the gene encoding HLA, and the effector molecule of the AGR system (RNA III) [70].

Conveniently, the drug does not cause the death of this commensal bacterium. It is more prone to inhibit the expression of virulence factors (the production of toxins, biofilm and motility structures) instead of killing them, leaving the skin liable to suffer dysbiosis and favoring the growth of more virulent microorganisms [71,72].

*Trans*-resveratrol can diminish the production of biofilms and toxins and reduce this bacteria’s motility. It also avoids the production of exotoxins (including the staphylococcal enterotoxin A) that can activate the immune response of T lymphocytes, which produce immunoglobulin E, increasing irritation and inflammation, and aggravating the clinical condition [66,73].

In addition, *trans*-resveratrol is a phenolic compound not belonging to the corticosteroids class, which implies some overtime administration advantages. Corticosteroids can lead to a higher susceptibility to skin infections (including acne), skin atrophy, striae, and hirsutism [74]. On the other hand, *trans*-resveratrol is generally considered safe for use, having fewer and milder side effects reported over the years [2]; thus, being suitable for treating chronic diseases such as atopic dermatitis, and may be helpful in achieving a healthy and healed skin without the side effects observed in more conventional drugs.

The films can act more to prevent the opportunistic infection caused by this bacterium than curing cases where the staphylococcal infection is already triggered. More studies would be needed to evaluate the effect of a concentration of 48 µg of *trans*-resveratrol released by PVP-SD-1:9 on the inhibition of *S. aureus* virulence factors.

The results of this study are quite promising, as the developed systems can sustain the controlled drug delivery as an anti-inflammatory medicine. In the case of PVP-SD-1:3, it can simultaneously treat inflammation and act as an antimicrobial agent to treat atopic dermatitis.

## 5. Future Perspectives

Considering that solid dispersions can increase the amount of drug-loaded into polymeric films, they can be further explored as ingredients to develop more potent drug delivery systems, allow new drugs to be incorporated into medicines, and increase their release and bioavailability. New therapies can be achieved for drugs that cannot be incorporated sufficiently enough to obtain the required dose-response for dermatological effects or other administration routes.

Another important finding is that spray drying technology seemed to maintain the chitosan present in the solid dispersions ionized and cause more interactions between this material and PVP. It caused more thinning and reinforcement of the films compared with the controls with chitosan in nature (not ionized), indicating that solid dispersions can enhance the films’ properties.

These findings show that polymeric films and solid dispersions are an exciting combination. Although solid dispersions cannot be used as a skin drug delivery system, polymeric films sustain controlled release to skin because they have increased bioadhesion and time of contact.

Finally, using a non-corticosteroid drug is promising due to the advantages of *trans*-resveratrol, which does not cause classic side effects and is a possible option for the basal treatment of atopic dermatitis without compromising the individual’s skin. It not only treats the inflammation, but also prevents the expression of virulence factors of opportunistic pathogens and may keep the balance of the skin’s microbiome, which can decrease side effects and antimicrobial resistance.

## 6. Conclusions

PVP films incorporated with *trans*-resveratrol: chitosan amorphous solid dispersions were successfully developed. The films were homogeneous, with barrier properties able to cover and adhere to inflamed skin to deliver the *trans*-resveratrol in a controlled manner with no cytotoxicity and an anti-inflammatory action. They also showed inhibitory effects against the opportunistic pathogen *S. aureus* (which could worsen the health condition).

The amorphous solid dispersions incorporated into the PVP films showed a promising strategy for maintaining the drug without recrystallizing with no incompatibilities among the formula components. The PVP and chitosan association enhanced the bioadhesive properties and promoted the sustained drug release, also permeating slowly through the skin, which indicates the ability of the drug to stay long enough in the skin layers to cause the pharmacological effect.

Finally, the association of amorphous solid dispersions and polymeric films can overcome both systems’ issues. Depending on the polymer, the drug can easily recrystallize when incorporated into polymeric films; however, amorphous solid dispersions enable the amorphization of the drug and may assure its release and bioavailability.

On the other hand, when using amorphous solid dispersions alone, they generally cannot achieve controlled drug delivery because of the lack of consistency and bioadhesive properties being necessary for their incorporation into other dosage forms; in the case of cutaneous administration, the polymeric films are ideal for this purpose.

## Figures and Tables

**Figure 1 pharmaceutics-14-01149-f001:**
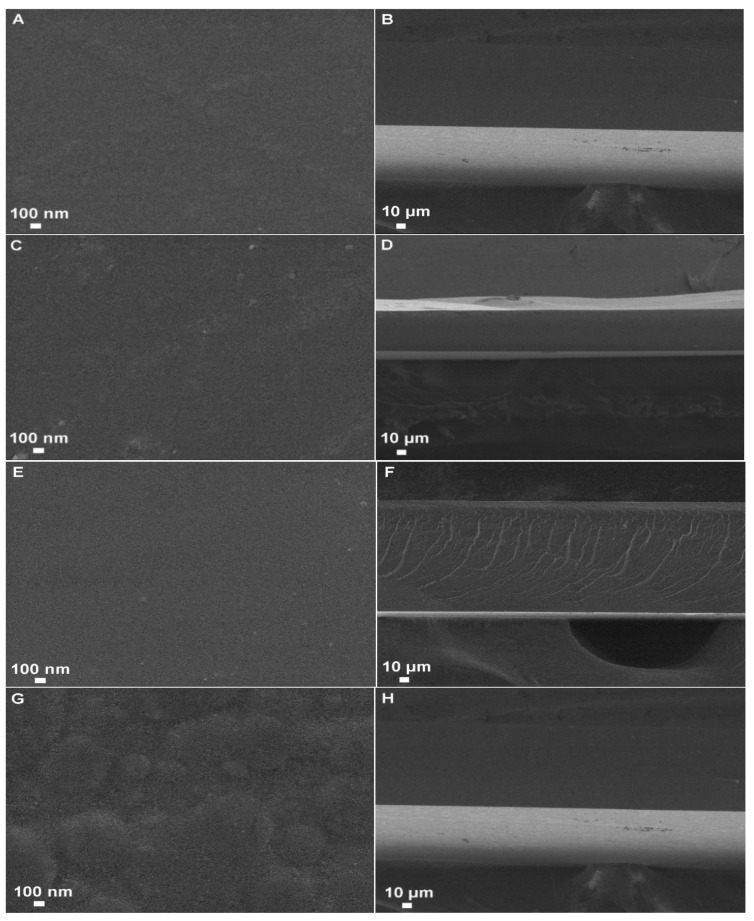
SEM images of the film’s surface (**A**,**C**,**E**,**G**) and cross-sections (**B**,**D**,**F**,**H**). **A**,**B**: C-PVP-1:9; **C**,**D**: C-PVP-1:3; **E**,**F**: PVP-SD-1:9; **G**,**H**: PVP-SD-1:3.

**Figure 2 pharmaceutics-14-01149-f002:**
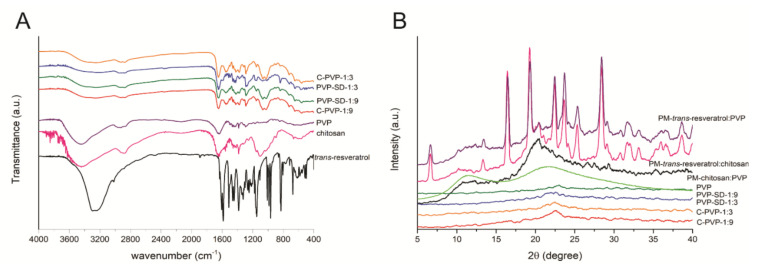
Physicochemical properties of raw materials, physical mixtures (PM), and membranes: (**A**) FTIR spectra; (**B**) DRX diffractograms.

**Figure 3 pharmaceutics-14-01149-f003:**
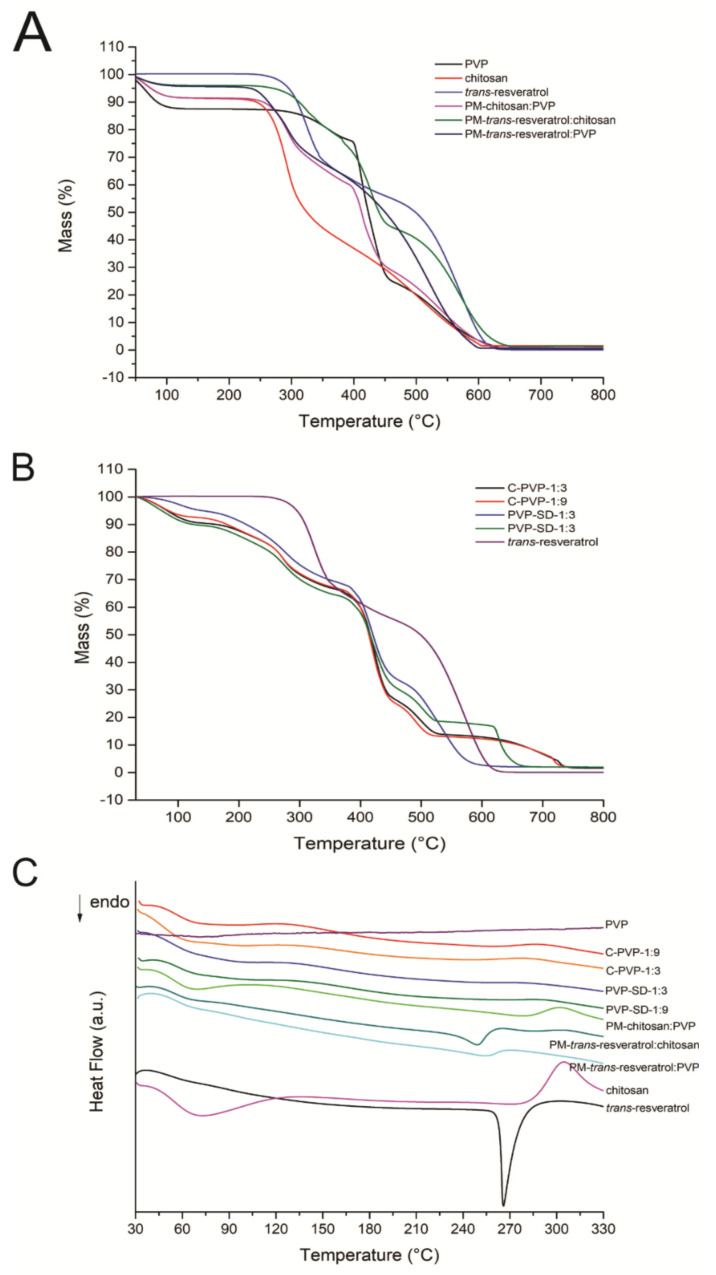
TG (**A**,**B**) and DSC (**C**) thermograms of the raw materials, physical mixtures (PM), and membranes.

**Figure 4 pharmaceutics-14-01149-f004:**
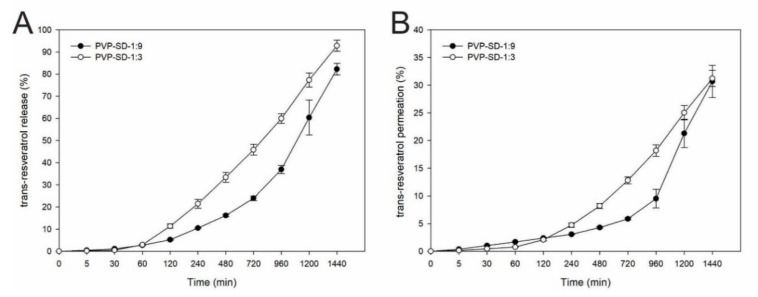
In vitro release (**A**) and in vitro cutaneous permeation (**B**) profiles of *trans*-resveratrol from the membranes. The profiles are presented as mean ± SD; n = 6.

**Table 1 pharmaceutics-14-01149-t001:** Solid dispersion and PVP films samples.

Formulation	*Trans*-Resveratrol:Chitosan Ratio	Chitosan (g)	TPGS (%)
SD-1:9	1:9	-	1
SD-1:3	1:3	-	1
PVP-SD-1:9	1:9	-	1
PVP-SD-1:3	1:3	-	1
C-PVP-1:9	-	0.90	1
C-PVP-1:3	-	0.75	1

Footnote: SD: solid dispersions. Control samples (C-) did not contain the *trans*-resveratrol.

**Table 2 pharmaceutics-14-01149-t002:** Grades of Cytotoxicity (ISO 10993-5:2009).

Grade (Cytotoxicity)	Cytotoxicity Zone
0 (absent)	No signal of discoloration under the sample area
1 (light)	Discoloration zone only under the sample area
2 (mild)	Discoloration up to 0.5 cm beyond the sample area
3 (moderated)	Discoloration between 0.5 and 1.0 cm beyond the sample area
4 (severe)	Discoloration greater than 1.0 cm beyond the sample area

**Table 3 pharmaceutics-14-01149-t003:** Thickness, liquid uptake ability at equilibrium (120 min), water vapor permeability (WVP), and mechanical properties of PVP films. Tests were performed in triplicate (mean ± SD).

Samples	Thickness (μm)	Liquid Uptake (%)	*WVP* (×10^−5^ g mm m^−2^ h^−1^ Pa^−1^)	*P_S_* (Mpa)	*E_b_* (%)	*P_E_* (kJ m^−3^)
C-PVP-1:9	157 ± 5.39 ^a^	1579.59 ± 31.16 ^a^	2.07	30.36 ± 2.05 ^a^	13.41 ± 2.45 ^a^	3.22 ± 0.647 ^a^
C-PVP-1:3	163 ± 8.35 ^a^	760.78 ± 52.64 ^b^	2.10	22.99 ± 5.37 ^a,c^	12.70 ± 3.78 ^a^	1.69 ± 0.790 ^b^
PVP-SD-1:9	123 ± 5.18 ^b^	983.94 ± 49.90 ^c^	1.38	32.20 ± 3.88 ^b^	15.43 ± 6.93 ^a^	4.24 ± 1.11 ^a,c^
PVP-SD-1:3	122 ± 4.74 ^b^	606.69 ± 27.10 ^b,c^	1.43	39.25 ±2.67 ^a,b^	14.72 ± 3.18 ^a^	5.28 ± 1.77 ^a,b,c^

Equal letters indicate no statistically significant difference between the samples (Test-t, *p* > 0.05). WVP = water vapor transmission, Ps = puncture strength, Eb = elongation at the break, Pe = perforation energy.

**Table 4 pharmaceutics-14-01149-t004:** Thermal data from the TG-DSC curves.

Sample	Step 1			Step 2			Step 3			Step 4			Step 5	
	T_Range_ (°C)	T_peak_ (°C)	∆m (%)	T_Range_ (°C)	T_peak_ (°C)	∆m (%)	T_Range_ (°C)	T_peak_ (°C)	∆m (%)	T_Range_ (°C)	T_peak_ (°C)	∆m (%)	T_Range_ (°C)	T_peak_ (°C)
C-PVP-1:9	30–105	↓75	6.66	105–335	↓206 ↑278	26.44	360–450	↓415	39.95	450–520	↑495	16.66	520–745	↑720
C-PVP-1:3	30–120	↓82	8.97	120–370	↑274	25.32	370–445	↓415 ↑442	37.30	445–530	↑505	14.96	530–750	↑730
PVP-SD-1:9	30–140	↓75	10.35	140–370	↑280	25.94	370–460	↓415 ↑440	33.33	460–535	↑505	12.01	535–700	↑730
PVP-SD-1:3	30–120	↓100	4.17	120–380	↑290	28.25	380–460	↑345	33.74	460–640	↑535 ↑548	31.87	----	----
Chitosan	30–126	↓73	8.47	149–310	↑300	36.40	310–607	↑325 ↑485	53.62	----	----	----	----	----
PVP	30–135	↓75	12.45	240–400	↓400	11.98	400–455	↓440	49.22	455–650	↑480	26.23	----	----
*Trans*- resveratrol	242–350	↓266	31.11	350–650	↑565	68.84	----	----	----	----	----	----	----	----
Chitosan: PVP	30–129	↓70	8.43	215–393	↑380	32.35	393–450	↓398 ↓435	28.75	450–625	↑485	29.88	----	----
*Trans*-resveratrol: PVP	30–120	↓80	3.90	240–380	↓265	19.03	380–450	↓430	30.44	450–650	↑560	45.23	----	----
*Trans*-resveratrol: chitosan	30–120	↓65	4.20	195–330	↑255 ↑265 ↑310	24.94	330–600	↑525	70.46	----	----	----	----	----

↓ Endo; ↑ Exothermic.

**Table 5 pharmaceutics-14-01149-t005:** Bioadhesion strength of the films in porcine skins. The same letters indicate no statistical difference between the samples (T-test, *p* > 0.05).

Formulations	Mean ± SD (N)
C-PVP-1:9	0.1373 ± 0.2240 ^a^
C-PVP-1:3	0.0629 ± 0.0155 ^b^
PVP-SD-1:9	0.1142 ± 0.0171 ^a,c^
PVP-SD-1:3	0.0999 ± 0.0115 ^d^

**Table 6 pharmaceutics-14-01149-t006:** *Trans*-resveratrol retention in skin layers after 24 h.

		PVP-SD-1:9	PVP-SD-1:3
*Stratum corneum*	Accumulative concentration (μg/cm^2^)	47.14 ± 53.41	189.56 ± 65.10
	Retained drug (%)	35.10 ± 4.87	27.29 ± 4.48
Dermis and epidermis layers (below the *stratum corneum*)	Accumulative concentration (μg/cm^2^)	59.06 ± 25.82	117.26 ± 14.53
	Retained drug (%)	18.18 ± 10.28	15.59 ± 1.42

**Table 7 pharmaceutics-14-01149-t007:** Qualitative results of cytotoxicity study.

Films	Halo (cm)	Cytotoxicity Degree
Control (+)	0.9	Severe
Control (−)	0	Absent
*Trans*-resveratrol	0.5	Moderate
C-PVP-1:9	0	Absent
C-PVP-1:3	0	Absent
PVP-SD-1:9	0.4	Light
PVP-SD-1:3	0.1	Light

**Table 8 pharmaceutics-14-01149-t008:** Inflammation and inflammation reduction results. Equal letters indicate no statistically significant difference between the samples (Tuckey test, *p* > 0.05).

Groups	Mean Inflammation (%)	Reduction in Inflammation (%)
*Trans*-resveratrol 3 mg/mL	30.85 ± 4.29	69.15 ^a^
PVP-SD-1:9	56.89 ± 7.19	43.11 ^a^
PVP-SD-1:3	33.98 ± 7.09	66.02 ^a^
C-PVP-1:9	80.37 ± 8.04	19.63 ^b,e^
C-PVP-1:3	83.58 ± 21.54	15.42 ^c,e^
Control + (dexamethasone 1 mg/g)	24.82 ± 6.96	75.18 ^d^
Control - (no treatment)	98.45 ± 7.21	1.55 ^e^

**Table 9 pharmaceutics-14-01149-t009:** Mean of the values (mm) of the inhibition zone of the samples (One-way ANOVA with Tukey’s post-test, *p* < 0.001).

Experimental Group	Mean ± SD
C-PVP-1:3	NI
C-PVP-1:9	NI
PVP-SD-1:3	4.11 ± 1.53
PVP-SD-1:9	NI
ETH-1:3	NI
ETH-1:9	NI
ETH	NI
NC	NI

NI: not inhibited; ETH: ethanol; NC: negative control.

## Data Availability

Not applicable.

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
