# Peer review of "Solid Dispersions Incorporated into PVP Films for the Controlled Release of Trans-Resveratrol: Development, Physicochemical and In Vitro Characterizations and In Vivo Cutaneous Anti-Inflammatory Evaluation"

_pharmaceutics, 2022, doi:10.3390/pharmaceutics14061149_

Round 1
Reviewer 1 Report
This article is interesting, useful, and well prepared.
Introduction it is in line with the Instructions for the Authors. The methodology is also corresponding to the experimental part points of interest. Results are representative and comprehensive with the remark already done in the General comment.
Discussion is also appropriate, but a more precise direction of future research should add more value.
References are in line with the scientific demonstration of the main issues.
Some necessary revisions are list below:
- Please review the equations that are not aligned on the page.
- Why you use only Staphylococcus aureus for microbial strain assay?
- What is the antimicrobial mechanism of the investigated samples?
- From the picture of the Petri dish with the size of the halos it does not seem to be an average of 4 mm ... the inhibition is almost non-existent. Please post other clearer pictures showing antibacterial inhibition. Eventually repeat the test.
Best regards!
Author Response
Dear reviewer,
Please see the the attachment.
Kind Regards,
Prof. Bruno Riccio

Reviewer 2 Report
The research article “Solid Dispersions Incorporated into PVP Films for the Controlled Release of Trans-Resveratrol: Development, Physicochemical and in vitro Characterizations and in vivo Cutaneous Anti-Inflammatory Evaluation” by Bruno Riccio et al aims to the develop the system which is suitable as a drug delivery system and capable to treat simultaneously the inflammation and infections related to atopic dermatitis. Authors have addressed the important problem of Trans-Resveratrol solubility and bioavailability. This is a timely article and the results presented are well supported.
I am enthusiastic about this article and supportive of its publication. I only offer some minor suggestions to improve readability and enhance the message of the paper (adopting them is optional).
Minor issues:
- The image quality of Fig 2, 3 could be improved.
- Figure 4: Authors should mention in figure legend: n and specify whether mean + sd or sem is plotted. Also, it would be useful to mention which statistical method is used.
- Authors could support their results and conclusion with the illustration showing the different formulations and its application.
Author Response

(The authors gave the same response as above.)

Round 2
Reviewer 1 Report
Accept in present form
Author Response
Dear Academic editor,
Please see the attachment.
Best regards,
Prof. Bruno Riccio
